# Enhancement of Mechanical Properties and Bonding Properties of Flake-Zinc-Powder-Modified Epoxy Resin Composites

**DOI:** 10.3390/polym14235323

**Published:** 2022-12-05

**Authors:** Xu Luo, Yu Li, Shuaijie Li, Xin Liu

**Affiliations:** Department of Basic, Naval University of Engineering, Wuhan 430033, China

**Keywords:** flake zinc powder, epoxy, modification, toughen

## Abstract

As a typical brittle material, epoxy resin cannot meet its application requirements in specific fields by only considering a single toughening method. In this paper, the effects of carboxyl-terminated polybutylene adipate (CTPBA) and zinc powder on the mechanical properties, adhesion properties, thermodynamic properties and medium resistance of epoxy resin were studied. A silane coupling agent (KH-550) was used to modify zinc powder. It was found that KH-550 could significantly improve the mechanical properties and bonding properties of epoxy resin, and the modification effect of flake zinc powder (f-Zn) was significantly better than that of spherical zinc powder (s-Zn). When the addition amount of f-Zn was 5 phr, the tensile shear strength and peel strength of the composites reached a maximum value of 13.16 MPa and 0.124 kN/m, respectively, which were 15.95% and 55% higher than those without filler. The tensile strength and impact strength reached a maximum value of 43.09 MPa and 7.09 kJ/m^2^, respectively, which were 40.54% and 91.11% higher than those without filler. This study provides scientific support for the preparation of f-Zn-modified epoxy resin.

## 1. Introduction

Epoxy resin has the advantages of excellent mechanical properties, a high bonding strength, low curing shrinkage, good corrosion resistance and a low price. It is widely used in the preparation of adhesives, composite materials and coatings [1,2,3]. However, cured pure epoxy resin has the defects of high brittleness and low fracture energy [4,5], which limits its development in the field of advanced materials. Toughening modifications of epoxy resin can effectively improve its toughness and other shortcomings [6,7,8].

The common methods of toughening modifications of epoxy resin include adding rubber elastomer, thermoplastic resin, inorganic nanoparticles and flexible segments [9,10,11]. Rubber-elastomer-toughened epoxy resin generally reacts with the epoxy group in the epoxy resin to form a block polymer through the active end group of the rubber. During the curing process, the rubber chain segment precipitates from the system to form a two-phase structure, which improves its comprehensive mechanical properties [12,13]. Studies have shown that the addition of rubber generally leads to a decrease in the heat resistance and glass transition temperature of the material [14,15]. Thermoplastic-resin-modified epoxy resin could maintain sufficient toughness while having good heat resistance, and could overcome the shortcomings of the poor heat resistance of rubber elastomers. The toughening principle is mainly the crack-riveting mechanism, that is, when the system is subjected to external force, the thermoplastic particles uniformly dispersed in the system can absorb certain amounts of fracture energy and prevent the diffusion of cracks, but the amount of thermoplastic resin used for toughening is large, the solubility is poor, the viscosity of the system is large, and there are some difficulties in the preparation process [9]. The toughening mechanism of inorganic nanoparticles is mainly in the crack-deflection mechanism, that is, when the system is subjected to external force, the well-dispersed inorganic nanoparticles deflect the crack to consume the fracture energy, so that the toughness is effectively improved [16], but the poor dispersion of nanoparticles is easy to agglomerate in the resin matrix, thus affecting the performance of the cured product. The flexible segment toughening epoxy resin mechanism occurs where the molecular design of the flexible segment is introduced into the curing agent or resin matrix, in the curing process of microscopic phase separation, causing the formation of two-phase structure, so as to improve the toughness of epoxy resin, impact resistance, etc., but this method with rubber elastomer toughening will appear with an elastic modulus, mechanical strength and heat resistance, and as such, the reduction of its application in structural materials is limited [17,18,19].

As a typical brittle material, epoxy resin cannot meet the application requirements of specific fields by only considering a single toughening method. The synergistic modification of epoxy resin can achieve the purpose of toughening and thermal performance improvement [14,20,21]. In the 1990s, linear polyester was applied to the toughening modification of epoxy resin, and the toughness, impact strength and tensile shear strength were improved. However, it belonged to the flexible chain segment and its structure was soft. When the addition amount was too small, the toughening effect was not obvious. When the addition amount was too large, although the toughness of the resin could be effectively improved, its tensile strength and heat resistance were lost. Therefore, in order to achieve the purpose of toughening epoxy resin with flexible segments and improve other properties, such as the mechanical properties and bonding properties, it was necessary to strictly control the added number of flexible segments. Therefore, a feasible way was to introduce another modifier (synergistic toughening) into the flexible-segment-modified epoxy resin matrix [22].

Polymer composites not only inherit a series of advantages from polymers, but also have the advantages of a reinforcing phase, which have been widely used in the flame retardant, anticorrosion and electromagnetic fields [23,24,25,26]. In recent years, due to the particularity of its structure, lamellar structural fillers have been increasingly applied in the field of anticorrosion. The commonly used lamellar fillers mainly include flake zinc powder, graphene, hydrotalcite, montmorillonite, etc. [27,28]. The flake zinc powder is commonly used in Dacromet technology and inorganic zinc-rich coatings. The prepared coatings have excellent anticorrosion performance. Due to its unique flake structure, flake zinc powder can be in close contact with the matrix and lap in parallel, so that the path of corrosive substances in the air or medium to diffuse into the interior is more tortuous and the distance is longer. At the same time, zinc powder can be used as a sacrificial anode in the corrosion process to protect the steel of the cathode, and finally form a dense protective layer to block the invasion of the corrosive medium. The flake zinc powder without surface treatment has the disadvantages of high surface energy and poor dispersion. In the actual application process, the phenomenon of uneven dispersion and uneven surface will occur, which seriously affects the corrosion resistance of the coating.

In this paper, bisphenol A epoxy resin (E-51) was used as the research object, phenolic amine (T-31) was used as the curing agent, epoxy-terminated polybutylene adipate (E-CTPBA) was used as the toughening agent (self-made in the laboratory, adding 20 phr, where phr represents the amount of E-51 needed to add per 100 phr; in this paper phr is used as the standard; for the E-CTPBA preparation method see the literature [29]) and zinc powder was used as a filler. The effects of different amounts of zinc powder on the mechanical properties, bonding properties, thermodynamic properties and medium resistance of carboxyl-terminated polyester/epoxy resin (CTPBA/EP) composites were systematically studied. The synergistic toughening mechanism of KH-550-modified flake zinc powder (f-Zn) and CTPBA was discussed, and the synergistic effects of improving the mechanical properties of materials was revealed.

## 2. Experimental Methods

### 2.1. Materials

Bisphenol A epoxy resin (E-51) with an epoxy value of 0.48~0.52, industrial grade, was purchased from Nantong Xingchen Synthetic Materials Co., Ltd., Nantong, China; Phenolic amine curing agent (T-31): industrial grade, amine value of 520 mg/KOH/g, was purchased from Guangzhou Yueqiang Chemical Co., Ltd., Guangzhou, China. Adipic acid (≥99.5%), an analytical reagent, was purchased from Tianjin Kemiou Chemical Reagent Co., Ltd. (Tianjin, China). The analytical reagent 1,4-butanediol (≥99.0%) was purchased from Xilong Scientific Co., Ltd., Shantou, China. P-toluenesulfonic acid (≥99.0%), an analytical reagent, was purchased from Tianjin Damao Chemical Reagent Factory, Tianjin, China. Triphenylphosphine (PPh_3_, ≥99.0%), an analytical reagent, was purchased from Aladdin Reagent Co., Ltd., Shanghai, China. Spherical zinc powder (s-Zn) and flake zinc powder (f-Zn): industrial grade, particle size of 400 mesh, was purchased from Tianjin Renmai Metal Materials Co., Ltd., Tianjin, China. Silane coupling agent (KH-550, ≥99%) was purchased from Kangjin New Material Technology Co., Ltd., Dongguan, China.

### 2.2. Modification of Zinc Powder

KH-550 was used to modify f-Zn and s-Zn. The KH-550-modified f-Zn was named Mf-Zn, and the KH-550-modified s-Zn was named Ms-Zn. The f-Zn and s-Zn were placed in an oven at 120 °C for 1 day and taken out for later use. Then, 9 wt.% KH-550 of zinc powder mass was weighed, and an appropriate amount of deionized water was added to prepare an aqueous solution with a mass concentration of 5%. In order to accelerate the hydrolysis, the pH was adjusted to 47, and placed in an ultrasonic cleaner. The solution was shaken for 30 min to fully hydrolyze the silane coupling agent; the measured f-Zn powder and s-Zn powder were added into the ethanol solution to prepare the ethanol solution with a mass concentration of 5 wt.% and shaken in an ultrasonic cleaner for 30 min to allow the zinc powder to mix evenly in the ethanol solution. The hydrolyzed silane coupling agent solution was added to the zinc powder ethanol solution and placed in a magnetic stirrer for modification. The reaction was stopped after magnetic stirring at 70 °C for 2 h. The modified zinc powder was suction-filtered and washed with anhydrous ethanol to remove the residual silane coupling agent. After repeated washing 5 times, it was dried in an oven at 120 °C and ground for later use.

### 2.3. Preparation of Zn/CTPBA/EP Epoxy Resin

A carboxyl-terminated polybutylene adipate (CTPBA) with a molecular weight of 1000 was prepared using the method of our laboratory, and epoxy-terminated polyester (E-CTPBA) was prepared by being epoxy-terminated at both ends. The resin matrix prepared by mixing 20 phr E-CTPBA and 100 phr E-51 was named CTPBA/EP. The detailed preparation method is set out in the literature [29].

Mf-Zn and Ms-Zn were added to CTPBA/EP in concentrations of 0 phr, 5 phr, 10 phr, 15 phr, 20 phr and 30 phr (0%, 5%, 10%, 15%, 20% and 30% of the mass of E-51), respectively. The high-speed mixer was fully stirred for 15 min. After 30 min of ultrasonic vibration, it was used as component A, and the amount of T-31 curing agent was 25 phr, which was used as component B. Mf-Zn-modified epoxy resin composites (Mf-Zn/CTPBA/EP) and Ms-Zn-powder-modified epoxy resin composites (Ms-Zn/CTPBA/EP) were prepared by mixing the two components, and all samples were vacuum defoamed and cured at 25 °C for 2 days. The preparation process is shown in Figure 1, and the formula design is shown in Table 1.

### 2.4. Characterization and Performance Testing

The tensile properties and impact properties were tested according to GB/T 2567-2008. The tensile properties were tested at 25 °C at a speed of 10 mm/min.

The impact properties were tested by non-notched impact. The plastic pendulum impact testing machine (PTM1000) was used for testing. The sample size was 120 mm × 15 mm × 10 mm.

The tensile shear strength test refers to GB/T 7124-2008, and the SUS321 stainless steel sheet size was 100 mm × 25 mm × 1.6 mm. A single lap method was used. The length of the bonding surface was 12.5 ± 0.25 mm, the width was 25 ± 0.25 mm and the thickness of the adhesive layer was about 0.2 mm. The overflow adhesive was cleaned in time. After 2 days of curing, the universal tensile testing machine was used to test at 25 °C and 2 mm/min.

The peel strength test refers to GB/T 2791-1995, and flexible material selected was stainless steel, with a size of 200 mm × 25 mm × 0.15 mm, 150 mm coating length and 100 mm/min speed stripping.

A Fourier transform infrared spectrometer (FT-IR, Thermo Scientific, Nicolet iS20, Waltham, MA, USA) was used to characterize the synthesized zinc powder before and after modification. The scanning range was 4000–400 cm^−1^, the resolution was 4 cm^−1^, and the scanning number was 32 times.

The contact angle of zinc powder before and after modification was measured by the contact angle measurement instrument SDC-80. The test solution was deionized water. The test method was the static drop method and the dosage was 3 μL each time.

The thermal stability of the modified epoxy resin was tested by a thermos gravimetric analyzer (HT/1600, Mettler Toledo, Shenzhen, China). About 10 mg of the sample was weighed in the crucible, and the heating range was 25–700 °C, the heating rate was 10 °C/min and the gas atmosphere was N_2_.

The glass transition temperature of the sample was measured by a dynamic thermomechanical analyzer (DMA1, Mettler Toledo). The measurement mode was a single cantilever, the heating rate was 3 °C/min, the temperature range was 25–150 °C and the test frequency was 1 Hz.

The scanning electron microscope (SEM, TESCAN MIRA LMS) was used to test the morphology of Zn and the impact fracture of the cured substance. After gold-plating on the fracture surface, the accelerating voltage was 15 keV.

An X-ray diffractometer (XRD, model: Rigaku SmartLab SE) was used to characterize the zinc powder. Before the test, the zinc powder was dried in an oven at 120 °C for 1 h to remove moisture, and then the zinc powder was placed in a clean slide and flattened. The test was performed by compaction with another slide, with a voltage of 40 kV, a current of 30 mA, a scanning rate of 2°/min and a scanning angle of 2θ ranging from 5° to 90°. The light source was Cu-Kα ray, and the test target was a copper target.

According to GB/T 1690-2010, the sample size was 25 mm × 50 mm × 2 mm. The sample was placed in an oven at 104 °C for 1 h, and after weighing, it was soaked in the medium (tap water, seawater, 10% HCl, 10% NaOH and diesel oil) to maintain a constant temperature of 25 °C. The mass W_0_ (accurate to 0.0001 g) before soaking was weighed by a precision electronic balance within 1 min. After soaking for a period of time, the sample was taken out, and the liquid on the surface of the sample was quickly wiped off with filter paper. The mass W_1_ after soaking was weighed (accurate to 0.0001 g), and the weighing was also completed within 1 min. The mass after soaking for t days was W_t_, and the mass change rate after soaking for t days was calculated according to Equation (1):(1)$$C_{t}=\frac{W_{t}{\;-W}_{0}}{W_{0}}\;\times \;100$$

C_t_ was the mass change rate of the sample (%), W_0_ was the initial mass before soaking (g), W_t_ was the mass of the sample after soaking (g), and the result was the median value of the three sample values. According to GB/T 13353-92, the tensile shear strength samples of adhesives soaked in different media for 7 days were tested, and the strength change rate ∆δ (decrease, %) was calculated as Equation (2), where Δδ was the change rate of strength (%), δ_0_ and δ_1_ were the arithmetic mean values of the tensile shear strength in the blank test and after 7 days of immersion in the medium, respectively.
(2)$$\Delta \delta =\frac{\delta_{0}{\;-\;\delta }_{1}}{\delta_{0}}\;\times \;100$$

## 3. Results and Discussion

### 3.1. Characterization of Modified Zinc Powder

The infrared spectra of s-Zn, Ms-Zn (KH-550-modified s-Zn), f-Zn and Mf-Zn (KH-550-modified f-Zn) were measured after grinding and tableting with KBr, as shown in Figure 2. For s-Zn and f-Zn without KH-550 modification, there was a strong O–H stretching vibration absorption peak at a wave number of 3433 cm^−1^, which could be attributed to the presence of adsorbed water. There was a weak C–H absorption peak at 2925 cm^−1^, which might be due to the attachment of oil during the preparation process. The Si–O–Si antisymmetric stretching vibration characteristic absorption peaks of Ms-Zn and Mf-Zn appeared at a wave number of 1089 cm^−1^. In addition, the Zn–O characteristic absorption peaks appeared at 560 cm^−1^ and their peak positions migrated after modification. KH-550 had a Si–O bending vibration absorption peak at 460 cm^−1^, which also appeared in Ms-Zn and Mf-Zn. The above analysis and the following means of representation proved that KH-550 modified s-Zn and f-Zn successfully.

The phases of f-Zn, Mf-Zn, s-Zn and Ms-Zn were analyzed by an X-ray diffractometer (XRD). The patterns are shown in Figure 3. It can be seen from the diagram that the main peak of the main phase Zn appeared at 2θ = 43.52°, corresponding to the (1 0 1) crystal plane, and the secondary peaks appeared at 2θ = 36.48° and 39.18°, respectively, which were the (0 0 2) and (1 0 0) crystal planes, thus determining that the zinc powder was a closely packed hexagonal structure. However, the characteristic absorption peaks of XRD spectra of f-Zn and s-Zn before and after modification did not move, so it was concluded that the modification of zinc powder by KH-550 did not change its crystal structure.

Scanning electron microscopy (SEM) is a common characterization method for microscopic morphology analysis. In order to observe the morphological characteristics of zinc powder before and after modification and analyze its surface elements, SEM and an energy dispersive spectrometer (EDS) were used to observe. The SEM images of f-Zn and Mf-Zn are shown in Figure 4a,c, respectively. Both f-Zn and Mf-Zn had a lamellar stacking structure, and the surface contained a small number of particles. However, the surface of Mf-Zn (Figure 4b) was rougher and there were more small particles attached to it. The energy spectra of Figure 4b,d showed that the content of carbon and oxygen in f-Zn particles was low, while the surface particles of Mf-Zn contained zinc, oxygen, carbon and silicon elements. The content of carbon and oxygen elements was greatly improved compared with that of unmodified Mf-Zn. The newly added silicon element came from KH-550. It showed that the silane coupling agent KH-550 had bonded with the –OH on the surface of f-Zn after hydrolysis, which proved that the modification of f-Zn was successful.

Similarly, Figure 4e,g show the SEM images of unmodified s-Zn and Ms-Zn, respectively. Both s-Zn and Ms-Zn were spherical, and the surface of s-Zn was relatively smooth, while the surface of Ms-Zn was coated with a large number of particles. The energy spectra in Figure 4f,h showed that the content of carbon and oxygen in the particles of s-Zn was low, while the chemical composition of the particles on the surface of Ms-Zn was zinc, oxygen, carbon and a small amount of silicon. The content of carbon and oxygen was greatly improved compared with the unmodified, and the newly added silicon element came from KH-550. It indicated that the silane coupling agent KH-550 was bonded with -OH on the surface of s-Zn after hydrolysis, which proved that the modification of f-Zn was successful.

Contact angle was an important criterion to judge the quality of filler modification. The contact angle before and after modification of zinc powder is shown in Figure 5a,b. It can be seen that the contact angle of s-Zn was 121.404°, while the contact angle of Ms-Zn was 128.484°. It had relatively good hydrophobicity and enhanced lipophilicity, which improved the compatibility with epoxy resin matrix and enhanced the comprehensive mechanical properties. Similarly, it can be seen from Figure 5c,d that the contact angle of f-Zn was 141.778°, while the contact angle of Mf-Zn was 150.104°, which was larger than that of s-Zn and Mf-Zn. The contact angle indicated that it had relatively good hydrophobicity, stronger lipophilicity and better compatibility with the epoxy resin matrix, which enhanced the comprehensive mechanical properties. The reason for this phenomenon might be that the silane coupling agent contained polar groups and non-polar groups. The polar groups were easily grafted on the surface of the filler by reacting with the hydroxyl groups contained in the zinc powder, while the other end was a non-polar group, which was wrapped on the surface of the zinc powder and can effectively improve its hydrophobicity. The above phenomena indicate that KH-550 modified zinc powder successfully.

### 3.2. Effect of Coupling Agent on Material Properties

The combination of filler and resin matrix can directly affect the performance of modified epoxy resin. Therefore, the effects of zinc powder treated with a coupling agent on the mechanical properties of carboxyl-terminated-polyester-modified epoxy resin was investigated. In this experiment, the addition of unmodified zinc powder (s-Zn and f-Zn) and KH-550-modified zinc powder (Ms-Zn and Mf-Zn) was 5 phr. The experimental results of the bonding strength and mechanical properties of the modified epoxy resin adhesive with the coupling agent treated and untreated fillers are shown in Table 2. It can be seen from the table that the adhesive properties, elongation at break and impact strength of the epoxy resin adhesive with unmodified s-Zn decreased, and the tensile strength increased to a certain extent. The peel strength, tensile strength and impact strength of KH-550-modified Ms-Zn/CTPBA/EP were improved compared with the pure CTPBA/EP resin matrix, and the tensile shear strength decreased, but all were higher than s-Zn/CTPBA/EP.

The adhesive properties and elongation at break of the epoxy resin adhesive (f-Zn/CTPBA/EP) with unmodified f-Zn decreased, and the tensile strength and impact strength increased to a certain extent. The mechanical properties and adhesive properties of epoxy resin adhesive (Mf-Zn/CTPBA/EP) with KH-550-modified f-Zn were improved compared with the pure CTPBA/EP matrix, and the values were higher than those of unmodified f-Zn. Relatively speaking, the bonding properties and mechanical properties of Mf-Zn/CTPBA/EP were better than those of Ms-Zn/CTPBA/EP. The above results showed that KH-550 had a certain promoting effect on the performance of the epoxy resin adhesive.

The reason for the above phenomenon might be that the specific surface area of unmodified zinc powder was large, the surface was rich in hydroxyl groups, and the compatibility between inorganic particles and organic matter was poor, which was not easy to diffuse in the resin matrix, resulting in agglomeration, and a large number of defects appearing in the curing process, which reduced the comprehensive mechanical properties. After the zinc powder was modified by the silane coupling agent KH-550, the improvement of the comprehensive performance might be due to the particularity of the structure of the silane coupling agent, which could couple the zinc powder with the resin matrix, inhibit the tendency of the filler agglomeration, improve the interface bonding strength, improve the compatibility between the two, and effectively improve the comprehensive performance of the epoxy resin.

### 3.3. Effect of Zinc Powder on Bonding Properties of CTPBA/EP

As shown in Figure 6, the tensile shear strength of Mf-Zn/CTPBA/EP showed a trend of increasing first and then decreasing. When the addition amount of Mf-Zn was 5 phr, the tensile shear strength reached a maximum value of 13.16 MPa, which was 15.95% higher than that without any addition. The tensile shear strength of Ms-Zn/CTPBA/EP epoxy resin adhesive was lower than that of the pure CTPBA/EP resin matrix, and increased slightly with the increase in the amount of Ms-Zn, but it was smaller than that of the pure CTPBA/EP resin matrix. This might be because Mf-Zn had good dispersibility when the content was low. Mf-Zn had a lamellar structure. The presence of KH-550 on the surface made it more hydrophobic, and its compatibility with epoxy resin was increased. It was easy to bond with the CTPBA/EP resin matrix to form a cross-linked network structure, which made the interface bonding strength higher and played a role in dispersing stress when it was subjected to external forces. The addition of Mf-Zn continued to increase, and the tensile shear strength began to decrease. This may be due to the fact that Mf-Zn has great surface energy, and excessive Mf-Zn can easily lead to poor dispersion and agglomeration. Stress concentration occurred at the interface of the Mf-Zn/CTPBA/EP epoxy resin, resulting in a decrease in bonding performance.

As shown in Figure 7, the effects of different amounts of Mf-Zn and Ms-Zn on the peel strength of CTPBA/EP were studied. When Mf-Zn or Ms-Zn was added, the peel strength increased first and then decreased, and the peel strength of Mf-Zn/CTPBA/EP was higher than that of Ms-Zn/CTPBA/EP. When the addition amount was 5 phr, the peel strength reached a maximum of 0.124 kN/m, which was 55% higher than that of the pure CTPBA/EP resin matrix. This may be due to the good dispersion of Mf-Zn at low concentrations. After being modified by KH-550, it was easy to bond with the CTPBA/EP resin matrix to form a cross-linked network structure, which played a role in dispersing stress when it was subjected to external force, and improved the interfacial bonding strength, toughness and ductility. The addition of Mf-Zn and Ms-Zn continued to increase, and the peel strength began to decrease. This may be due to the agglomeration caused by the excessive dispersion of the filler, and the stress concentration occurred at the interface of Zn/CTPBA/EP, resulting in a decrease in peel strength.

### 3.4. Effect of Zinc Powder on Mechanical Properties of CTPBA/EP

The effect of the addition of Mf-Zn and Ms-Zn on the tensile strength and elongation at break of the CTPBA/EP epoxy resin is shown in Figure 8. The increase in zinc powder caused the tensile strength of Mf-Zn/CTPBA/EP and Ms-Zn/CTPBA/EP to increase first and then decrease, but the modification effect of Mf-Zn was more significant. When the addition amount was 5 phr, the maximum value was 43.09 MPa, which was 40.54% higher than that without Mf-Zn. The elongation at break of Mf-Zn/CTPBA/EP and Ms-Zn/CTPBA/EP decreased with the increase in the addition amount, but the elongation at break of Mf-Zn/CTPBA/EP was higher than that of Ms-Zn/CTPBA/EP. The reason for the above phenomenon may be that the zinc powder could form a good interface connection with the resin matrix after the coupling agent treatment. When it was subjected to tensile action, the stress could be well-dispersed through the interface. In addition, the strength of the particles was large, and the fracture needed to consume more energy. Because of its good adhesion with the resin matrix, it can prevent the expansion of microcracks, so that it needed more stress to pull out from the resin matrix, which was macroscopically manifested as the enhancement of tensile strength, that is, the so-called crazing-riveting mechanism. When zinc powder was added, the elongation at break decreased significantly, which may be due to the addition of flake zinc powder to reduce the spatial distance between the cross-linked chains to below the critical length, so that the fracture performance decreased [30]. When the addition amount was too great, due to the increased density between the filler particles, it was easy to cause the agglomeration phenomenon. When subjected to external force, stress concentration was easy to occur, which reduced the tensile strength and elongation at break.

The effect of Mf-Zn and Ms-Zn addition on the impact strength of CTPBA/EP is shown in Figure 9. It can be seen from the diagram that the addition of a small amount of Mf-Zn or Ms-Zn can effectively improve the impact strength of the material, and with the increase in the addition amount, the impact strength increased first and then decreased, and the modification effect of Mf-Zn was significantly better than that of Ms-Zn. When the content of Mf-Zn was 5 phr, the impact strength reached a maximum of 7.09 kJ/m^2^, which was 91.11% higher than that without filler. This may be because when the amount of zinc powder added was less, it could be more evenly dispersed in the resin matrix, and the contact surface area with the epoxy resin was larger, which could improve the interfacial bonding between the nanoparticles and the resin matrix. At the same time, the filler could absorb a certain amount of energy when impacted, hinder the crack propagation of the resin and prevent it from developing into a destructive fracture, so as to achieve the purpose of toughening [11]. With the increase in the addition amount, the impact strength of Mf-Zn/CTPBA/EP and Ms-Zn/CTPBA/EP began to decrease. This may be due to the increase in interface defects and agglomeration when the amount of filler was too great. When it was subjected to external forces, stress concentration was easy to occur, which made the performance of the composite system worse and led to a decrease in impact strength [5].

The SEM images of the impact fracture surface of CTPBA/EP cured samples with different additions of Mf-Zn are shown in Figure 10. When the amount of Mf-Zn powder added was low, as shown in Figure 10b,c, a large number of shear deformations, cavities, crack deflections and terminations can be observed, in which shear deformation can absorb part of the impact energy and improve the toughness; the formation of voids was due to the deintercalation behavior of Mf-Zn in the matrix. When the sample was subjected to impact force, part of Mf-Zn peeled off from the resin matrix, and absorbed a certain impact energy, which effectively improved its impact toughness. At the same time, it can be observed that Mf-Zn particles were evenly distributed in the resin matrix, and there were relatively few particles exposed to the outside, and some particles were located at the crack tip, which played a pinning effect, effectively inhibited the transmission of cracks and improved the toughness [31].

The energy dispersive spectrometer (EDS) was used to analyze and detect the elements of the surface exposed particles when the addition amount of Mf-Zn was 5 phr. As shown in Figure 11a, the elements of the exposed particles were mainly Zn, C, O and a small amount of Si, which proved that these particles were Mf-Zn. When the addition amount was 5 phr, the particles were dispersed more evenly, the exposed particles were less, the fracture surface was rough, the ductile fracture was more obvious, and the fracture needed to consume more energy, so as to improve the impact strength. When the addition of Mf-Zn continued to increase, a large number of exposed agglomerated particles can be observed, and the agglomeration became more obvious with the increase in the addition amount, thus reducing the impact strength [31]. Similarly, the SEM images of the impact fracture surface of Ms-Zn-modified CTPBA/EP are shown in Figure 12. The phenomenon and mechanism explanation were the same as those of Mf-Zn-modified CTPBA/EP. The EDS diagram of elemental analysis of exposed particles on the surface when the addition amount of Ms-Zn was 5 phr is shown in Figure 11b. Similar phenomena can be observed with Mf-Zn, and the phenomenon explanation was the same as Mf-Zn.

### 3.5. Thermal Stability

Because the mechanical properties and bonding properties of Mf-Zn/CTPBA/EP were better than Ms-Zn/CTPBA/EP, Mf-Zn was selected as the research object. Figure 13 shows the thermogravimetric (TG) curves and DTG curves of Mf-Zn/CTPBA/EP in a N_2_ atmosphere. From Figure 13a, it can be seen that the thermal weight loss process of epoxy resin modified by Mf-Zn was similar to that of epoxy resin matrix without Mf-Zn in a N_2_ atmosphere. The addition of Mf-Zn significantly improved the thermal stability of the CTPBA/EP resin matrix. The initial temperature of thermal weight loss (IDT) and residual carbon rate increased with the increase in Mf-Zn content. The reason for this phenomenon might be that Mf-Zn itself was an inorganic particle, and its thermal stability was good. After adding the system, it formed a strong interaction with the CTPBA/EP resin matrix, so that the energy required for fracture was higher, resulting in improved thermal stability. The increase in the residual carbon rate may be due to the higher proportion of inorganic fillers; the lower the weight loss rate, the higher the residual carbon rate.

Figure 13b is the DTG curve of Mf-Zn with different addition amounts. It can be seen from the diagram that the DTG curve of CTPBA/EP resin matrix had two peaks, which meant that the decomposition of the resin matrix was divided into two steps, namely, the thermal decomposition of the resin matrix and the oxidation of the pyrolysis residual carbon. The addition of Mf-Zn made the second peak become gentle, and the temperature T_max_ corresponding to the maximum thermal weight loss rate moved in the high temperature direction. The data are shown in Table 3. This may be due to the fact that Mf-Zn modified by a silane coupling agent was easy to bond with the resin matrix to form a cross-linked network structure, which made the firmness of the resin matrix larger, thereby improving the thermal stability of the Mf-Zn/CTPBA/EP epoxy resin. However, the T_max_ of the Mf-Zn/CTPBA/EP was basically unchanged with the increase in the amount of filler added, all at about 380 °C.

### 3.6. Dynamic Thermal Mechanical Properties

Figure 14 is the tanδ-T curve of CTPBA/EP modified by Mf-Zn with different addition amounts and Table 4 shows the damping data of CTPBA/EP modified by different contents of Mf-Zn. It can be seen that in the test temperature range, the material had only one tanδ peak, corresponding to only one glass transition temperature, indicating that the Mf-Zn/CTPBA/EP was thermodynamically completely compatible. When Mf-Zn was filled, the glass transition temperature (*Tg*) of the system increased. When the addition amount was 5 phr, the glass transition temperature increased rapidly from 70.3 to 75.6 °C, and the loss factor increased rapidly from 0.78 to 1.03. Appendix A was the E′ and E″ curves of CTPBA/EP epoxy resin modified with different Mf-Zn content. After that, *Tg* decreased with the increase in the addition amount, but it was still greater than that of pure CTPBA/EP. This may be because the addition of the flake filler affected the movement of the polymer molecular chain and improved its damping performance. The flaky fillers were modified by KH-550 to improve the compatibility with the epoxy resin matrix. Due to the van der Waals force, it can form a molecular chain entanglement with the surface of the epoxy resin matrix, and play the role of a chemical cross-linking point in the system, which limited the fluidity of the chain segment, thus increasing the glass transition temperature [32,33].

### 3.7. Medium Resistance

Figure 15a shows the mass change rate of the cured product of Mf-Zn-modified CTPBA/EP with the addition amount of 5 phr after immersion in acid, alkali, salt and other media. It can be seen from the figure that the order of the mass change of the cured product of Mf-Zn/CTPBA/EP was: NaOH > HCl > seawater > tap water > diesel. On the one hand, a large number of hydroxyl groups were produced after the curing of the epoxy resin and a small number of polar groups, such as the primary and secondary amino groups contained in the curing agent, were hydrophilic and could adsorb a small amount of water. After soaking in various media, the sample showed different degrees of swelling, increasing the quality; on the other hand, OH^−^ caused an easy hydrolysis reaction with ester group destruction of cross-linked structure so that there was a small amount of curing off, so that the quality was reduced; the two reacted together so that the quality of the sample changed. The mass change rate of Mf-Zn/CTPBA/EP in various media was low, and the solvent resistance was good. This may be because Mf-Zn had a unique lamellar structure, which can be in close contact between the zinc powder and the matrix. The parallel overlap made the path of the corrosive substance in the medium to the interior more tortuous and longer, so that the mass change was smaller.

Figure 15b shows the tensile shear strength of Mf-Zn/CTPBA/EP adhesive cured with 5 phr Mf-Zn-modified CTPBA/EP adhesive after soaking in acid, alkali, salt and other media. It can be seen from the graph that the influence of media on the tensile shear strength of the Mf-Zn/CTPBA/EP epoxy resin cured products was as follows: NaOH > HCl > sea water > tap water > diesel oil. It can be seen from Table 5 that the tensile shear strength of Mf-Zn/CTPBA/EP decreased less after immersion in the medium, and the medium resistance was better than that of CTPBA/EP. This might be due to the fact that Mf-Zn can improve the interface bonding between the filler and the resin matrix through the action of the silane coupling agent, so that the filler was uniformly dispersed in the epoxy resin, and the network structure formed after curing can effectively resist the penetration of the medium molecules. In addition, its unique sheet structure was overlapped in parallel in the resin, so that the path of the corrosive substance in the medium to the interior was more tortuous and the distance was longer. At the same time, Mf-Zn could be used as a sacrificial anode to protect the steel of the cathode during the corrosion process, and finally formed a dense protective layer to prevent the invasion of corrosive media.

## 4. Conclusions

A zinc-powder-modified epoxy resin was prepared by modifying a CTPBA/EP epoxy resin matrix with f-Zn and s-Zn, respectively. It had excellent bonding properties, mechanical properties and medium resistance. It had certain application value for adhesives that need to be used in harsh environments (such as seawater pipelines). Through the structure and morphology characterization of Mf-Zn and Ms-Zn and the bonding performance test, mechanical properties test, medium resistance analysis and impact section morphology characterization, the following conclusions were reached.

The tensile strength, peel strength, tensile shear strength, impact strength and elongation at break of Mf-Zn and Ms-Zn modified by KH-550 were better than those of unmodified f-Zn and s-Zn. Relatively speaking, the effect of Mf-Zn-modified epoxy resin adhesive was better than that of Ms-Zn. The addition of Mf-Zn significantly increased the tensile shear strength and peel strength of the epoxy resin adhesive, while the addition of Ms-Zn decreased the tensile shear strength. Relatively speaking, the bonding performance of Mf-Zn-modified epoxy resin adhesive was significantly better than that of Ms-Zn. When the addition amount of Mf-Zn was 5 phr, the tensile shear strength of Mf-Zn reached a maximum value of 13.16 MPa, which was 15.95% higher than that without the addition, and the peel strength reached 0.124 kN/m, which was 55% higher than that without the addition. The addition of Mf-Zn and Ms-Zn significantly improved the tensile strength and impact strength of epoxy resin adhesive, and the tensile strength and impact strength increased first and then decreased with the increase in addition amounts and the elongation at break decreased with the increase in addition amounts. The mechanical properties of Mf-Zn-modified epoxy resin were significantly better than that of Ms-Zn. When the addition amount of Mf-Zn was 5 phr, the tensile strength of Mf-Zn reached a maximum of 43.09 MPa, which was 40.54% higher than that without filler, and the impact strength reached a maximum of 7.09 kJ/m^2^, which was 91.11% higher than that without filler. The addition of Mf-Zn could improve the thermal stability of CTPBA/EP. The glass transition temperature and loss factor tanδ increased. When the addition of Mf-Zn was 5 phr, *Tg* reached 75.6 °C, which was 5.2 °C higher than that without filler, and tanδ increased from 0.78 to 1.03. Compared with CTPBA/EP, Mf-Zn/CTPBA/EP had better medium resistance.

## Figures and Tables

**Figure 1 polymers-14-05323-f001:**
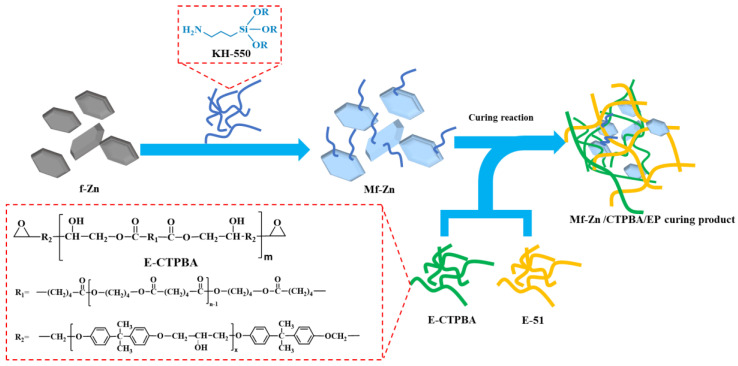
The schematic diagram of the preparation process of Mf-Zn/CTPBA/EP.

**Figure 2 polymers-14-05323-f002:**
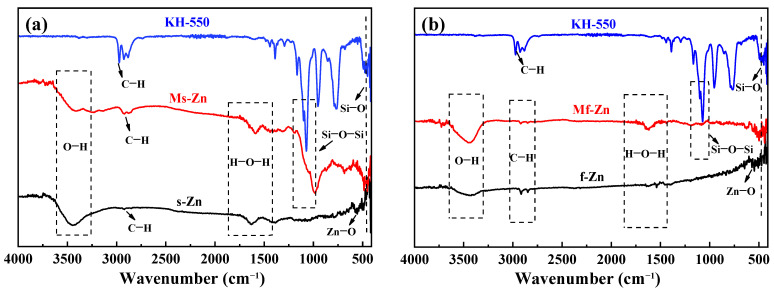
Infrared spectra of (**a**) s-Zn and Ms-Zn; (**b**) f-Zn and Mf-Zn.

**Figure 3 polymers-14-05323-f003:**
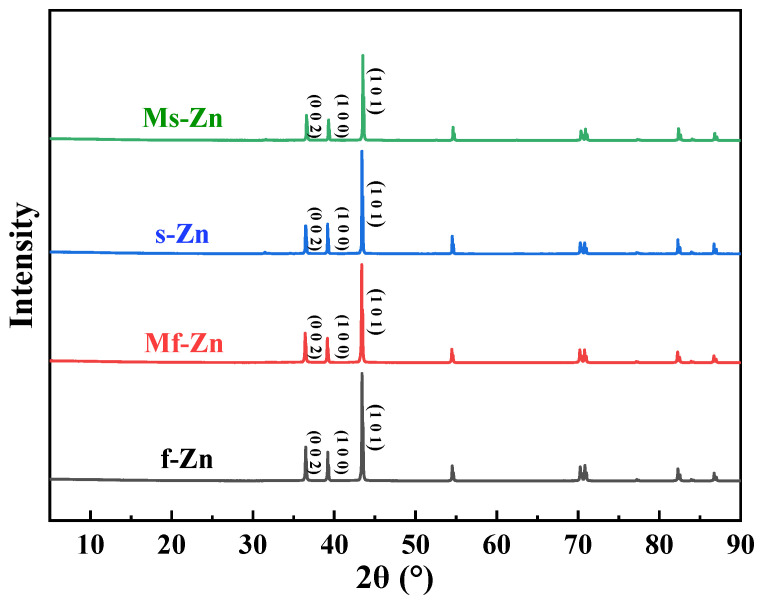
XRD spectra of f-Zn, Mf-Zn, s-Zn and Ms-Zn.

**Figure 4 polymers-14-05323-f004:**
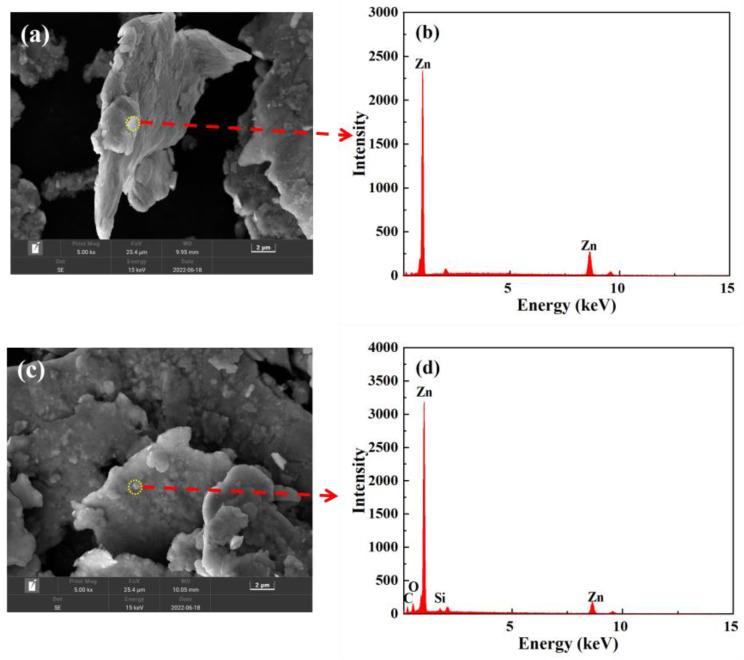
SEM images of (**a**) f-Zn, (**c**) Mf-Zn, (**e**) s-Zn and (**g**) Ms-Zn; The dashed line points to the EDS spectra of (**b**) f-Zn, (**d**) Mf-Zn, (**f**) s-Zn, (**h**) Ms-Zn for this location.

**Figure 5 polymers-14-05323-f005:**
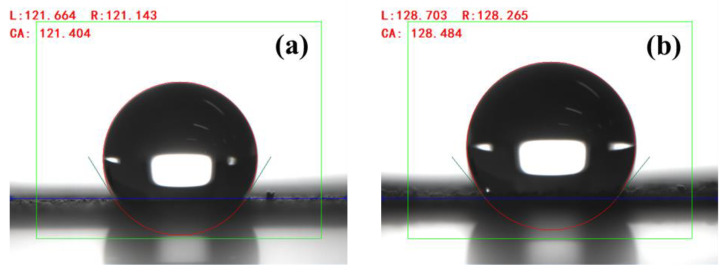
Contact angle of (**a**) s-Zn; (**b**) Ms-Zn; (**c**) f-Zn; (**d**) Mf-Zn.

**Figure 6 polymers-14-05323-f006:**
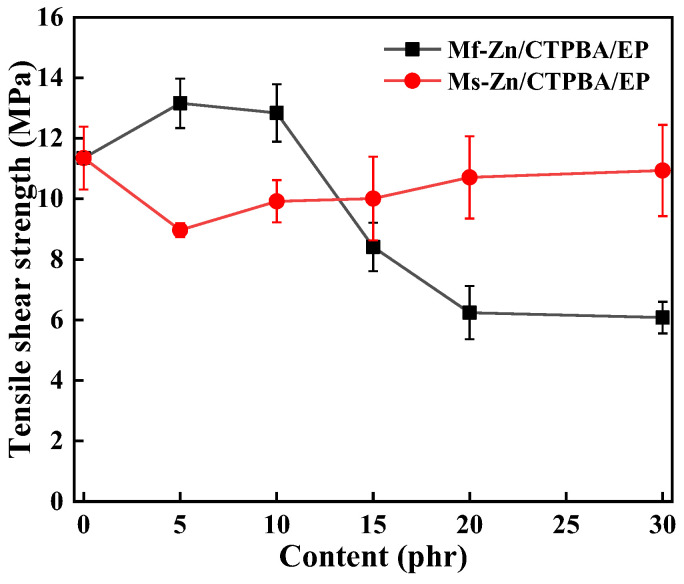
Effect of Mf-Zn and Ms-Zn content on the tensile shear strength of the CTPBA/EP epoxy resin.

**Figure 7 polymers-14-05323-f007:**
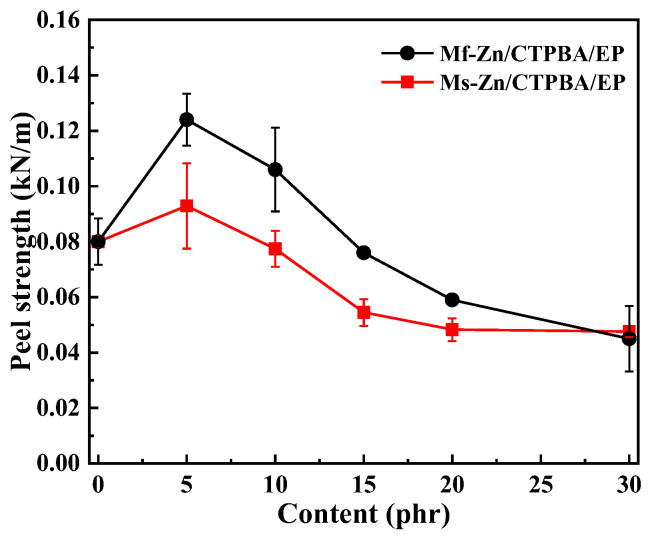
Effect of Mf-Zn and Ms-Zn content on the peel strength of the CTPBA/EP epoxy resin adhesive.

**Figure 8 polymers-14-05323-f008:**
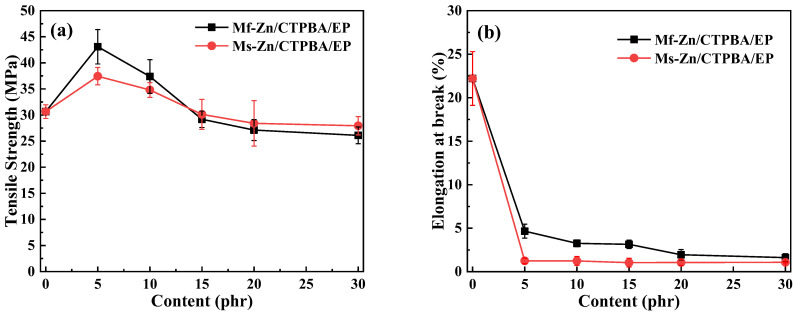
(**a**) Tensile strength and (**b**) elongation at break curve of Mf-Zn and Ms-Zn on the CTPBA/EP epoxy resin adhesives.

**Figure 9 polymers-14-05323-f009:**
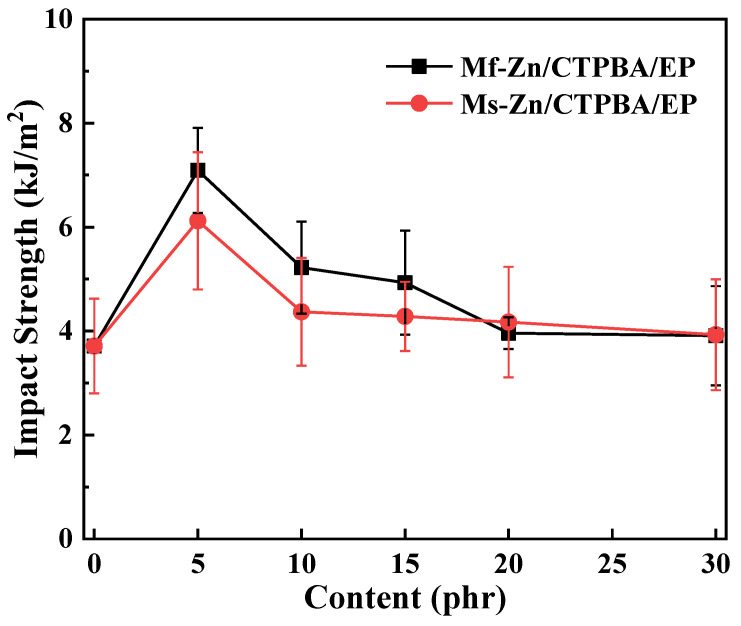
Impact strength curve of Mf-Zn and Ms-Zn on CTPBA/EP.

**Figure 10 polymers-14-05323-f010:**
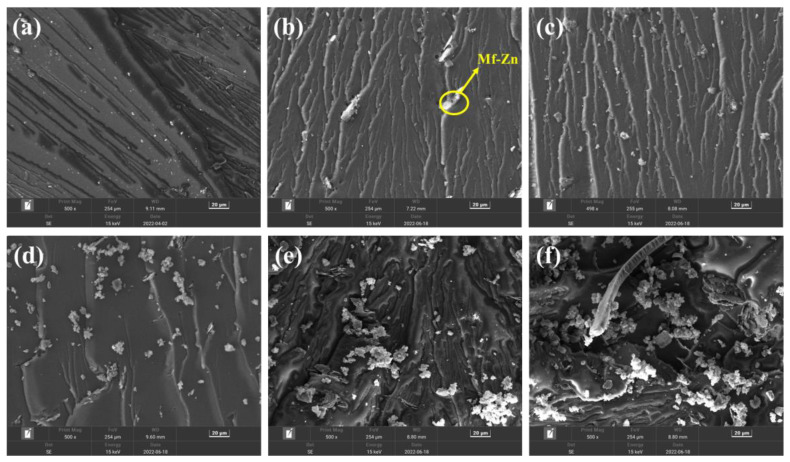
SEM images of (**a**) 0 phr; (**b**) 5 phr; (**c**) 10 phr; (**d**) 15 phr; (**e**) 20 phr and (**f**) 30 phr Mf-Zn-modified CTPBA/EP epoxy resin impact section.

**Figure 11 polymers-14-05323-f011:**
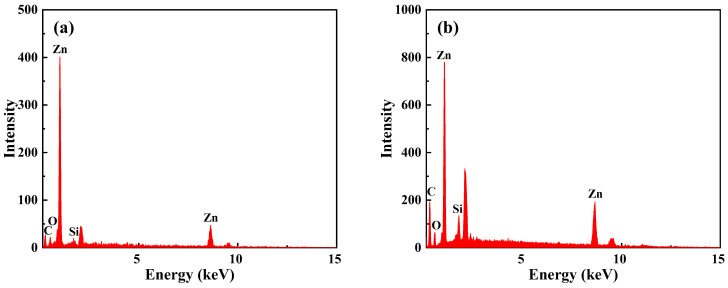
EDS diagram of impact section of epoxy resin adhesive with 5 phr (**a**) Mf-Zn, (**b**) Ms-Zn content.

**Figure 12 polymers-14-05323-f012:**
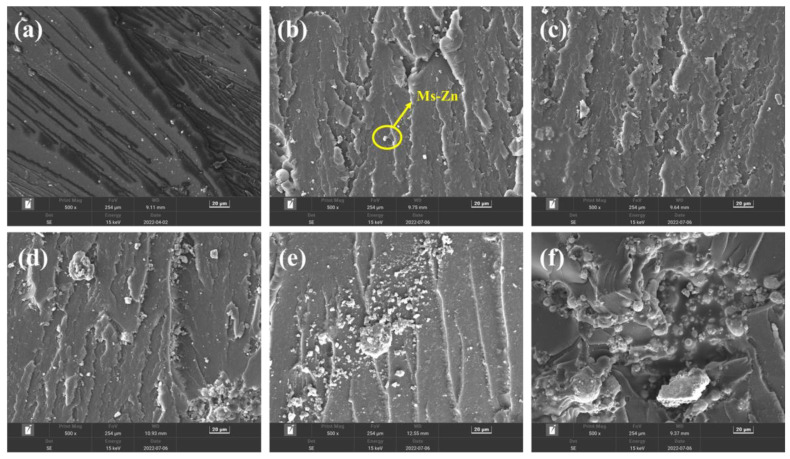
SEM images of (**a**) 0 phr; (**b**) 5 phr; (**c**) 10 phr; (**d**) 15 phr; (**e**) 20 phr and (**f**) 30 phr Ms-Zn-modified CTPBA/EP epoxy resin impact section.

**Figure 13 polymers-14-05323-f013:**
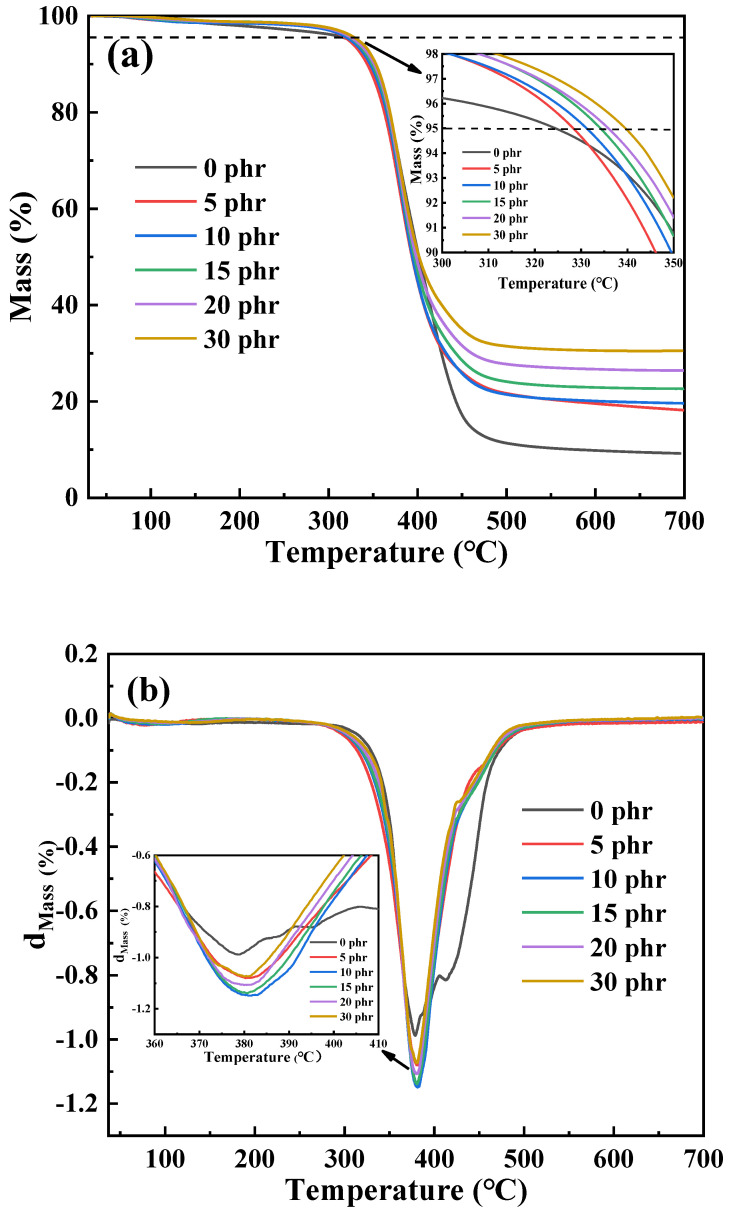
(**a**) TG curve; (**b**) DTG curve of CTPBA/EP epoxy resin adhesives modified with different Mf-Zn content.

**Figure 14 polymers-14-05323-f014:**
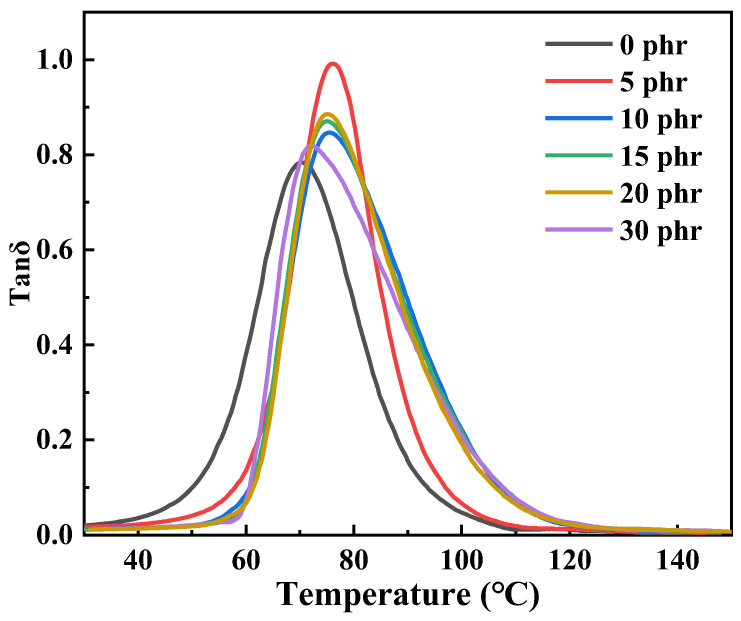
T-tanδ curves of CTPBA/EP epoxy resin adhesives modified with different Mf-Zn content.

**Figure 15 polymers-14-05323-f015:**
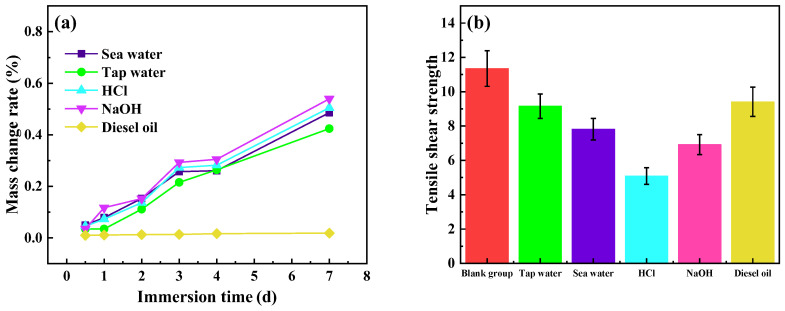
(**a**) The mass change rate of Mf-Zn/CTPBA/EP immersed in different media; (**b**) Tensile shear strength of Mf-Zn/CTPBA/EP immersed in different media.

**Table 1 polymers-14-05323-t001:** Formula of Zn/CTPBA/EP.

Sample	E-CTPBA (phr)	EP (phr)	T-31 (phr)	Ms-Zn (phr)	Mf-Zn (phr)
CTPBA/EP	20	100	25	0	0
Ms-Zn/CTPBA/EP-5	20	100	25	5	0
Ms-Zn/CTPBA/EP-10	20	100	25	10	0
Ms-Zn/CTPBA/EP-15	20	100	25	15	0
Ms-Zn/CTPBA/EP-20	20	100	25	20	0
Ms-Zn/CTPBA/EP-30	20	100	25	30	0
Mf-Zn/CTPBA/EP-5	20	100	25	0	5
Mf-Zn/CTPBA/EP-10	20	100	25	0	10
Mf-Zn/CTPBA/EP-15	20	100	25	0	15
Mf-Zn/CTPBA/EP-20	20	100	25	0	20
Mf-Zn/CTPBA/EP-30	20	100	25	0	30

**Table 2 polymers-14-05323-t002:** Effect of coupling agents on Zn/CTPBA/EP epoxy resin adhesive.

Sample	Tensile Shear Strength (MPa)	Peel Strength (kN/m)	Tensile Strength (MPa)	Impact Strength (kJ/m^2^)	Elongation at Break (%)
CTPBA/EP	11.35	0.08	30.66	3.71	22.21
s-Zn/CTPBA/EP	7.84	0.03	33.02	2.80	0.97
Ms-Zn/CTPBA/EP	8.97	0.09	37.44	6.12	1.25
f-Zn/CTPBA/EP	6.00	0.05	35.01	6.46	1.39
Mf-Zn/CTPBA/EP	13.16	0.12	43.09	7.09	4.66

**Table 3 polymers-14-05323-t003:** Thermal degradation parameters of the Mf-Zn/CTPBA/EP epoxy resin adhesive.

Sample	IDT (°C)	T_50_ (°C)	T_max_ (°C)	Residual Carbon Rate at 700 °C (%)
0 phr	324.39	401.50	378.80	9.19
5 phr	328.41	393.71	380.31	18.19
10 phr	331.47	394.19	381.32	19.62
15 phr	334.51	396.24	380.67	22.65
20 phr	336.25	398.32	380.05	26.42
30 phr	339.88	401.74	380.13	30.55

**Table 4 polymers-14-05323-t004:** Thermal degradation parameters of CTPBA/EP epoxy resin adhesives modified with different Mf-Zn content.

Content	*Tg*/°C	DTR/°C (tanδ ≥ 0.3)	Tan δ_max_
0 phr	70.3	26 (58–84)	0.78
5 phr	75.6	24 (65–89)	1.03
10 phr	75.4	31 (65–96)	0.85
15 phr	75.2	31 (65–96)	0.87
20 phr	74.7	30 (65–95)	0.89
30 phr	72.6	31 (64–95)	0.81

**Table 5 polymers-14-05323-t005:** The medium resistance of Mf-Zn/CTPBA/EP.

Medium	CTPBA/EP		Mf-Zn/CTPBA/EP	
	Tensile Shear Strength (MPa)	Change Rate of Strength (%)	Tensile Shear Strength (MPa)	Change Rate of Strength (%)
Blank group	11.35	0	13.16	0
Tap water	9.16	19.30	10.30	21.73
Sea water	7.82	31.10	9.69	26.37
HCl (10%)	5.09	55.15	8.71	33.81
NaOH (10%)	6.92	39.03	8.53	35.18
Diesel oil	9.41	17.09	11.66	11.40

## Data Availability

Not applicable.

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
