# Peer review of "Enhancement of Mechanical Properties and Bonding Properties of Flake-Zinc-Powder-Modified Epoxy Resin Composites"

_polymers, 2022, doi:10.3390/polym14235323_

Round 1

Reviewer 1 Report

In this manuscript, flake zinc powder was incorporated to modify the epoxy resin composites and the authors observed enhancement of mechanical and bonding properties. Overall, the experiments were well-designed and the manuscript was well-written. I support its publication after the following points are properly addressed.

1. At what temperature did the curing reaction take place and for how long?

2. How does the incorporation of zinc powder (either flake or sphere) affect the curing kinetics?

3. In the abstract and conclusion part, it was mentioned the dielectric properties were also studied. However, no dielectric experiments or results were discussed.

Reviewer 2 Report

Dear Authors,

In this paper authors modified the Zn filler by a silane coupling reagent and utilized as reinforcing filler for epoxy resin. The research methodology is fine and the results are informative. Paper is well written and I recommend some major revisions before final acceptance as suggested bellow.  

1.     Please modify these sentences The two components were mixed and fully mixed. Vacuum defoaming and sample preparation after full mixing. The Mf-Zn modified CTPBA / EP epoxy resin composite was named Mf-Zn/CTPBA/EP, and the Ms-Zn modified CTPBA/EP epoxy resin composite was named Ms-Zn/CTPBA/EP.

2.     The FT-IR of Silane coupling agent (KH-550) should be presented and have to see the peak shifts for a successful Zn-modification. I guess there should be a Zn-O-Si peak in the successful modification or at least peak shift. Si-O-Si peak can be obtained from remaining coupling reagent even after washing. Authors have to confirm.

3.     Generally, for reinforcing filler tan δ height becomes lower. Could you please provide loss modulus and storage modulus?

4.     Please include some suitable references in the mechanical properties discussion section. 

Round 2

Reviewer 2 Report

Thanks for your well revision.